# Artificial Neural Networks in Motion Analysis—Applications of Unsupervised and Heuristic Feature Selection Techniques

**DOI:** 10.3390/s20164581

**Published:** 2020-08-15

**Authors:** Marion Mundt, Arnd Koeppe, Franz Bamer, Sina David, Bernd Markert

**Affiliations:** 1Institute of General Mechanics, RWTH Aachen University, 52062 Aachen, Germany; mundt@iam.rwth-aachen.de (M.M.); koeppe@iam.rwth-aachen.de (A.K.); markert@iam.rwth-aachen.de (B.M.); 2Institute of Biomechanics and Orthopaedics, German Sport University Cologne, 50441 Cologne, Germany; S.David@dshs-koeln.de

**Keywords:** gait kinematics, inertial measurement unit, LSTM neural network, model order reduction, proper orthogonal decomposition

## Abstract

The use of machine learning to estimate joint angles from inertial sensors is a promising approach to in-field motion analysis. In this context, the simplification of the measurements by using a small number of sensors is of great interest. Neural networks have the opportunity to estimate joint angles from a sparse dataset, which enables the reduction of sensors necessary for the determination of all three-dimensional lower limb joint angles. Additionally, the dimensions of the problem can be simplified using principal component analysis. Training a long short-term memory neural network on the prediction of 3D lower limb joint angles based on inertial data showed that three sensors placed on the pelvis and both shanks are sufficient. The application of principal component analysis to the data of five sensors did not reveal improved results. The use of longer motion sequences compared to time-normalised gait cycles seems to be advantageous for the prediction accuracy, which bridges the gap to real-time applications of long short-term memory neural networks in the future.

## 1. Introduction

The analysis of human motion is of great interest to many different applications: it can be used to identify injury risk or to increase performance during sports-related tasks [1], but also in many clinical applications [2,3]. In particular, with regard to daily life, wearable technology has shown its feasibility in both sports [4] and clinical applications [5]. Using these systems might help to prevent injuries or the onset of motion related diseases. For this purpose, easy-to-use feedback systems are necessary to inform the user of potential risks [6,7].

In order to realise such systems in daily life, a certain level of accuracy is required to provide adequate feedback. The level of accuracy is highly dependent on the application and therefore needs to be evaluated for each particular research question. Lebleu et al. [8] stated an error of less than 5° to be acceptable for clinical gait analysis, but the errors reported using IMU systems to range between 5 and 18.8°. These inaccuracies regarding the determination of joint angles can be related to a drift of the signals over time, perturbations in the magnetometer readings and the calibration procedure necessary to align the sensor to the segment to set up the anatomical model [8,9,10,11].

To overcome these limitations, different approaches have been suggested. Besides manually aligning sensor and segment axes, functional calibration movements can be used for the alignment. The manual alignment assumes the segment and sensor axes to be parallel. For the functional calibration, the user needs to be able to execute the required position of movement. Expert knowledge is necessary for both methods to ensure a meaningful and repeatable calibration [8,12]. Lebleu et al. [8] achieved mean RMSE values for functional calibrations of 2.0 to 4.1°. Another approach is the exploitation of kinematic joint constraints to determine joint angles. This approach is for now limited to the determination of the 2D joint angles [13,14]. The use of machine learning, in particular, artificial neural networks, is gaining more and more relevance in biomechanical time-series estimation and has recently been reviewed by Gurchiek et al. [15]. They found hybrid approaches incorporating domain knowledge by feature selection into the model to be helpful for accurate predictions. Different approaches were used to automatically derive the most relevant information from the data and reduce redundancy: an exhaustive greedy algorithm [16] has been used to find the most relevant features to predict the centre of pressure trajectory [17] or the 3D ground reaction force using pressure insoles [18]. The correlation between different inputs has been determined to exclude highly correlated input features from 3D segment angles for the prediction of the knee adduction moment [19]. Ardestani et al. [20] performed a Fisher discriminant analysis to recognise relevant EMG signals and a partial correlation-kernel mutual information technique to select independent marker trajectories for the prediction of the knee contact force. Principal component analysis (PCA) was regularly used to reduce the number of features in EMG signals for the prediction of joint kinematics [21,22] or muscle force [23]. However, Chen et al. [24] presented deep belief networks to outperform PCA for feature selection in EMG signals. Oh et al. [25] used a self-organising map first to exclude highly correlated features from marker trajectories, and subsequently, a general regression neural network to find those inputs that were highly correlated to the outputs to finally predict the ground reaction force. To estimate running kinematics, Gholami et al. [26] used a forward sequential algorithm and a genetic algorithm to determine the best input features from strain data. Ziai and Menon [27] heuristically reduced the input parameters for the prediction of joint moments from EMG data and compared the results. They found an increased error for a reduced number of input channels. For the application of inertial sensors, the number and placement of sensors used was in general based on *a priori* decisions [28,29,30,31,32,33,34]. Shahabpoor et al. [35] used correlation techniques to find the optimum inertial sensor positions for the prediction of the ground reaction force. They achieved the highest accuracy using one sensor at the L5 vertebra and two sensors on the thighs.

However, none of the aforementioned studies analysed the influence of feature selection on the prediction by comparing the results to those without feature selection. The minimisation of the number of sensors necessary for getting accurate predictions is of high relevance because it simplifies the application during daily life and reduces the costs of wearable systems. Additionally, the use of fewer sensors will reduce the risk of misalignment. Errors in the sensors’ rotation especially cause inaccurate results [36]. For that reason, this study aimed to analyse which combination of sensors is favourable for the prediction of the 3D lower limb joint angles. Additionally, PCA was applied to the data. Thereby, the number of input features was reduced further while accounting for orientation errors. Since the amounts of inertial sensor data and the simultaneously measured marker-based data were limited, inertial sensor data were simulated based on optical motion capture data to enlarge the database. The validity of simulated data was presented and evaluated previously. The accuracy was $r_{pelvis}=0.95\pm 0.08$, $r_{right\;thigh}=0.88\pm 0.12$, $r_{left\;thigh}=0.91\pm 0.08$, $r_{right\;shank}=0.91\pm 0.11$, $r_{left\;shank}=0.92\pm 0.10$ [12]. Long short-term memory (LSTM) neural networks were trained on the different input data. We hypothesised that the use of three instead of five sensors placed on the pelvis and lower limbs would not decrease the prediction accuracy of the neural network significantly and that the application of a PCA would have a regularising effect on the network. This might encourage the use of less sensors and help to account for problems with regard to random orientation errors.

In the following Materials and Methods section an overview on the dataset is given, followed by a brief description of feature selection, LSTM neural networks and the data analysis performed. In the Results section, an overview of the contributions of single components to the principal component analysis is given, as is the prediction accuracy. Subsequently, the results are discussed, limitations are identified and a final conclusion is drawn.

## 2. Materials and Methods

An overview of the workflow is displayed in Figure 1 and detailed descriptions of each step are given in the following sections.

### 2.1. Data Set

The dataset used for this study pools data of different studies [10,12,37,38]. All data were collected at the German Sport University Cologne. The studies were approved by the Ethical Committee of the German Sport University Cologne and all participants provided their informed written consent. The dataset comprised 115 participants (50 females, 37.0 (18–75) years). A number of 24 participants underwent knee arthroplasty 1.8 ± 0.4 years post-surgery prior to gait analysis. Each participant executed level walking trials at self-selected speeds ranging from 0.8 to 2.0 $m/s^{-1}$. The motion was recorded using an optoelectronic motion capture system (VICONTM, MX F40, Oxford, UK, 100–125 Hz). The joint angles were calculated using a custom MATLAB script based on the recommendations of the International Society of Biomechanics [10,39]. The inertial sensor data—tri-axial linear acceleration and angular rate—were simulated as detailed in [12]. IMU data were simulated for the pelvis, both shanks and both thighs. The joint angles were calculated for the hip, knee and ankle joints of both legs in all three dimensions. For evaluation a five-fold cross-validation was performed. For this purpose, one fixed test set was split from the complete dataset, as were five different validation sets. The training set contained approximately 65% of the data, the validation set 15% and the test set 20%. The splits were undertaken randomly, ensuring that no overlapping between the sets occurred and no data of one participant were split between datasets (see Table 1).

### 2.2. Feature Selection

For feature selection, two different approaches have been performed. First, the number of (simulated) sensors was reduced manually. For this purpose four different LSTM neural networks were trained and their parameters were optimised for the best estimation of the joint angles:5 sensors: pelvis, right thigh, left thigh, right shank, left shank (PTS-net)3 sensors: pelvis, right thigh, left thigh (PT-net)3 sensors: pelvis, right shank, left shank (PS-net)1 sensor: pelvis (P-net).

Additionally, the dimensions of the PTS-net input data were reduced before optimising and training another LSTM network using the proper orthogonal decomposition (POD) [40,41]. The POD provides an optimal low-dimensional, uncorrelated description of a high-dimensional, correlated process (the gait dataset with *n* input vectors) using a set of *l* POD basis vectors. Hereby, the goal was to make the number of input vectors significantly smaller than the required number of POD basis vectors *n* (l≪n) so that a low-dimensional description of the high-dimensional complex systems could be possible. Without loss of generality, the transformation is expressed as:(1)$$\begin{array}{c} x\approx x\left(l\right)=\sum_{i=1}^{l}\Phi_{i}q_{i}=\Phi q\;. \end{array}$$

The problem is specified by the condition
(2)$$\begin{array}{c} E\left\{{∥x-x(l)∥}^{2}\right\}\leq E\left\{{∥x-\hat{x}\left(l\right)∥}^{2}\right\}\;, \end{array}$$
where x is identified as a random vector, which is, in our case, the measured data at a certain time instant. Condition (Equation 2) ensures that the POD approximation using *l* basis vectors x(l) is always better than the approximation using any other possible basis $\hat{x}\left(l\right)$. This leads to the objective function for the mean square error ε that depends on the chosen number of basis vectors included
(3)$$\begin{array}{c} \varepsilon {\left(l\right)}^{2}=E\left\{{∥x-x(l)∥}^{2}\right\}⟶\min \end{array}$$
subject to the orthonormality condition
(4)$$\begin{array}{c} \Phi_{i}\Phi_{j}=\delta_{ij}\;,\;i,j=1...l\;. \end{array}$$

The POD can be realised using the principal component analysis (PCA), the singular value decomposition and the Karhunen Loève decomposition. In this paper, we realise the POD using the PCA, which can be used to reduce a model with linear dependencies in the data [24]. In dynamical systems it is observed that indeed a surprisingly small number of POD basis vectors is sufficient for an accurate description of the full system [42]. In other words, the main features of the system can, generally, be described by a few degrees of freedom in a low-order subspace. Moreover, in this paper we must note that the first five principal components described about 95% of the variance of each sample. Hereby, the 30 features recorded from five IMU sensors were reduced to five features using PCA. Thereafter, sequences of these features of lengths varying from 100 to 1000 time steps constituted the inputs to the PCA-net.

### 2.3. Long Short-Term Memory Neural Network

A fully-connected feedforward neural network network layer *l* needs a large number of parameters $W_{l}$ and $b_{l}$, because the shape of the weight tensor $W_{l}$ scales quadratically with the dimensions of the layers. For time series, wherein it is common practice to flatten the sequence axis when using fully-connected networks, this means that the time dependency of the data is taken away, increasing the number of inputs. For this reason, it becomes difficult to find the most relevant time-dependent patterns in the data, despite the large number of parameters [43].

To overcome this, alternative, recurrent networks have been developed for the prediction of time series data and the preservation of time dependencies (Figure 2). While fully-connected neural networks need consistent sequence lengths as inputs and outputs, recurrent neural networks can learn from arbitrary sequence lengths. They process only the preceding time steps and not the complete sequence (Figure 3). In contrast to fully-connected neural networks, they have time-delayed inner recursions and inner states that serve as memory. Unfortunately, the memory of RNNs suffers from exploding or vanishing gradients during the back-propagation, which leads to distinct errors during the learning process. For detailed information on this, see (Goodfellow et al. [43], pp. 396–399).

One specific recurrent neural network is the long short-term memory neural network, which was introduced to overcome the long-term dependency problem [44,45]. The difference between an LSTM and a standard recurrent neural network is the presence of gate layers the LSTM, which consist of four different layers that decide how to update the cell state for the next recursion. The gate layers ensure that relevant information from the past can be memorised and information only relevant for a short time can be dismissed. Three different gates are used to update the cell state and thereby calculate the activation for the next recursion (Figure 4) [43].

For each sensor combination one LSTM neural network was trained. To achieve the best results, the architecture and hyperparameters of each network were optimised for the underlying dataset using a hyperband search [46]. For the PT-net, PS-net and PCA-net, the architecture after the hyperparameter search was 512-1024, while for the PTS-net and P-net the best architecture was 256-256. For all networks, the optimal learning rate was 3e-4 and the dropout rate 0.4.

### 2.4. Data Analysis

The correlation of single sensor readings to the five principal components used for training the PCA-net was evaluated.

To compare the prediction accuracy, the root mean squared error (RMSE) and correlation coefficient between the predicted and target values for all 18 joint angles were calculated. Since the test dataset that was analysed was chosen to be similar for all five models, a repeated measures ANOVA was calculated on the RMSE values to find differences between the single models. In case of differences, a post-hoc paired t-test with Bonferroni correction was evaluated.

## 3. Results

### 3.1. Principle Component Analysis

The PCA was calculated individually for each sample to reduce the model’s complexity and reduce it to its main features. The first five principal components of each sample described 94.6% of the variance in the data—42.9% described by the first, 22.6% by the second, 14.6% by the third, 10.0% by the fourth and 4.5% by the fifth. The analysis of the PCA loadings (Figure 5) revealed larger correlations between the sensor readings and the first two PCA components than for the last three. The strongest correlations between the sensor readings and the first PCA component were found for the pelvis angular rate around the medio-lateral (*z*) axis (*r* = −0.716) and vertical (*y*) axis (*r* = 0.746), both thigh sensors’ vertical (*y*) axes (right *r* = 0.847, left *r* = 0.777) and both shank sensors’ medio-lateral (*z*) accelerations (right *r* = 0.749, left *r* = −0.896). Additionally, the angular rate of the left shank around the anterior-posterior (*x*) axis (*r* = −0.861) and medio-lateral (*z*) axis (*r* = −0.823) showed strong correlations. For the right shank the correlations were only moderate (anterior-posterior *r* = 0.588, medio-lateral *r* = −0.660). In general, the angular rate readings showed a stronger correlation with the PCA components than the acceleration.

### 3.2. Prediction Accuracy

All networks showed good accuracy with correlation coefficients larger than 0.8, indicating strong correlations, and RMSE values smaller than 3°. The mean prediction accuracy was lowest for the PCA-net (r=0.826±0.066, RMSE=2.81±0.66) and P-net (r=0.879±0.050, RMSE=2.29±0.62), while the other networks showed similar accuracies—the PS-net being most accurate (PTS-net r=0.921±0.039, RMSE=1.71±0.55, PT-net r=0.918±0.041, RMSE=1.78±0.63, PS-net r=0.924±0.038, RMSE=1.60±0.57). Figure 6 and Figure 7 display single examples of the ground truth and predicted joint angles for the PCA-net and PS-net. In particular, in the sagittal plane, the agreement between the ground truth and predicted values is high, while it is distinctly lower in both other motion planes.

In Figure 8, the distributions of the correlation coefficients for all networks, motion planes, joints and the left and right side are depicted separately. The mean accuracy was similar for both sides. Hence, left and right were summarised for clarity in Table 2 and Table 3, which display the mean correlation coefficients and RMSE values. The prediction of the PCA-net showed a higher variance than the other networks. The sagittal joint angles were predicted with very high accuracy. Only the prediction of the ankle joint angle showed a lower accuracy using the PCA-net. In general, the deviations were larger for the prediction of the ankle joint angle than for the other joints. The variance in the prediction accuracy was higher in the non-sagittal motion planes than in the sagittal, main motion plane, although the mean correlation coefficient still indicated strong correlations for all angles besides the hip transverse plane. The largest RMSE values were found in the sagittal planes for the prediction using the PCA-net. For all other models, the RMSE was smaller in the sagittal plane than in the minor motion planes, not exceeding 3°, except for the prediction of the knee joint angle in the transverse plane using the P-net.

The repeated measures ANOVA showed differences between the RMSE values of the different networks (p<0.001). The post-hoc paired t-test with Bonferroni correction revealed significant differences (p<0.001) between all networks besides the PTS-net and PT-net.

## 4. Discussion

This study compared an unsupervised and a heuristic feature selection technique for the determination of joint angles during gait based on simulated inertial sensor data. As an unsupervised technique, PCA was chosen to reduce the input features from 30 sensor readings to five principle components. Regarding the heuristic approach, four different sensor combinations were investigated: (1) sensors placed on the pelvis, both thighs and both shanks, (2) sensors placed on the pelvis and both thighs, (3) sensors placed on the pelvis and both shanks and (4) a single sensor placed on the pelvis. With the heuristic approach a minimisation of the required physical number of sensors becomes possible, which improves the applicability of the system due to reduced costs and less effort for daily life usage. The PCA approach can reduce computational costs and can have an regularising effect by accounting for different sensor orientations.

The analysis of the RMSE values showed differences between all five networks trained besides the PTS-net and PT-net. The PS-net, i.e., the three-sensor configuration, outperformed the network reliant on all five sensors. This indicated that the use of five sensors is not necessary and underlines our initial statement that the use of five sensors is not advantageous over the use of three sensors only. These findings match the analysis of the PCA loadings (Figure 5). The strongest correlations between the sensor readings and the first five principle components could be found for the pelvis and shank sensors, while the correlation for the thigh sensors was lower. From a mechanical perspective, these results are not surprising. The pelvis sensor records the motion of the centre of mass—a common reference point in gait analysis. The shank sensors record the motion of the entire leg. The thigh sensor records the motion of only the upper leg with respect to the reference point. Hence, the combination of information on the pelvis and shank motion spans two of the three analysed joints, the hip and knee, while the information on the knee joint is not directly available from the thigh sensors. The use of only a single sensor placed at the pelvis revealed good results, especially with regard to the sagittal motion plane. For practical applications, wherein lower accuracy is acceptable in return for a simple measurement setup, the use of this single sensor might be a good compromise. Interestingly, the neural network trained on the first five principle components of the sensor data performed worst. As there was information taken away from the input data, no perfect accuracy could be expected, but as there was information of all 30 sensor readings in the data, we hypothesised the PCA-net would perform better than the neural networks based on less sensors. We assumed the PCA to have a regularising effect on the neural network. We could not find this in our results, but observed an increased variance in the accuracy over all joints and motion planes. This might be attributed to the application of a PCA on each sample instead of finding one set of principle components for all samples.

The prediction of the joint angles in the sagittal plane is generally very good for all networks, showing the smallest variance in the prediction accuracy. For the minor motion planes, the variance in prediction accuracy is increased, which might result in samples showing large errors (e.g., the knee frontal plane angle depicted in Figure 6). This can be attributed to the larger variance in these motion planes in general and the fact that smaller movements might be detected adequately by inertial sensors. If a sample shows a large deviation from the mean of the dataset, the prediction accuracy is decreased. By enlarging the dataset and the variance in the training data, the prediction accuracy might be improved.

This study was—to the authors’ knowledge—the first that used an LSTM neural network for the prediction of joint angles from data of complete motion sequences of arbitrary lengths instead of sequences that sdfd time-normalised to 100% stance phase or gait cycle. With this approach, preprocessing of the data needs less effort and the gap to real-time applications of neural networks becomes smaller. In a previous study based on the same dataset [12], we used a fully-connected feedforward neural network for the prediction of joint angles based on simulated IMU data for time-normalised gait phases. We achieved a mean correlation coefficient of r=0.87. The weakest correlation was found for the knee joint frontal plane angle. For the sagittal plane, all joint moments could be predicted with a correlation coefficient r>0.95. The mean RMSE value was 4.1°, not exceeding 6° [12]. In another study, we compared the use of a fully-connected feedforward network and an LSTM network on a smaller dataset with time-normalisation to gait cycles. The results of this study compared to those of the previous study are displayed in Table 4. The longer sequences seem to be advantageous for the prediction of the minor motion plane joint angles, but the sagittal plane correlation coefficients were slightly lower in the recent study. The RMSE was smaller compared to the previous study.

The use of machine learning for the prediction of motion kinematics based on raw IMU data is subject of current research. Hence, the comparison to other studies is difficult. Using kinematic constraints, Seel et al. [47] achieved RMSE values of 3.3° and 1.6° for the prediction of sagittal knee and ankle joint angles. These values are higher than those achieved in this study. Based on the common approach using Kalman Filters [48] for sensor fusion, accuracy in a similar range to this study could be achieved [49,50] using more sensors. Whether the achieved accuracy is sufficient is highly dependent on the application and needs to be carefully evaluated. As the accuracy is in the same range as the accuracy of commercial systems, the application of machine learning methods for this task should be considered, because it has the major advantage that the anatomical model—which most measurement deviations in inertial sensor-based joint angle calculations can be attributed to [10]—is intrinsically learned. Hence, no calibration movements, postures or very accurate sensor-to-segment alignment are necessary, if the training dataset is large enough and shows sufficient variance [12].

The major limitation of the current study was the use of simulated IMU data only. Further validation based on a measured IMU dataset should be undertaken in future work. Nevertheless, previous studies showed good agreement of simulated and measured data, and a comparable prediction accuracy of neural networks on both. Hence, we assume that the use of measured data resulted in a comparable level of accuracy [12,51,52].

In future work it might be worthwhile to investigate the use of sensors placed on the feet. Thereby, all three joints would be covered, which might improve the prediction accuracy. On the other hand, the additional degrees of freedom in this model could also limit its prediction accuracy. Additionally, the use of PCA for the data should be further evaluated. If the PCA could be performed online instead of during the preprocessing, the information of the complete dataset could be covered and lead to more general principal components which might improve the usability of this approach.

## 5. Conclusions

The use of machine learning to estimate joint angles from inertial sensors is a promising approach to in-field motion analysis. Since no distinct physical connection between the input and output data is necessary, neural networks have the opportunity to estimate joint angles from a sparse dataset. This enables the reduction of sensors necessary for the determination of all three-dimensional lower limb joint angles. The use of longer sequences of motion compared to time-normalised gait cycles seems to be advantageous for the prediction accuracy, which bridges the gap to real-time applications of LSTM networks in future work.

## Figures and Tables

**Figure 1 sensors-20-04581-f001:**
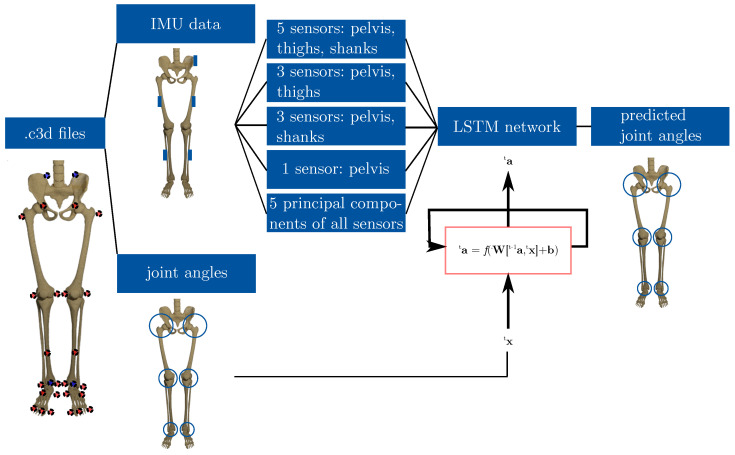
Scheme of the workflow. The marker trajectories of gait trials, saved as .c3d files, were used to calculated inertial sensor data and the joint angles of the lower limbs. The inertial data were divided into five subsets and used as input to train long short-term memory neural networks that were used to predict the joint angles. To adapt its weights and biases for the prediction, the network received the calculated joint angles as target values.

**Figure 2 sensors-20-04581-f002:**
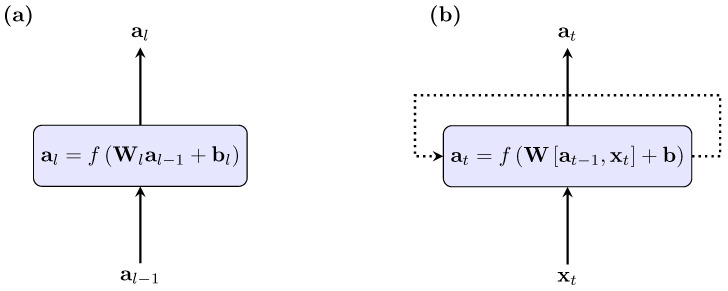
A standard fully-connected feedforward neural network (**a**) and a recurrent neural network (**b**). The recurrent neural network is extended by an additional loop.

**Figure 3 sensors-20-04581-f003:**
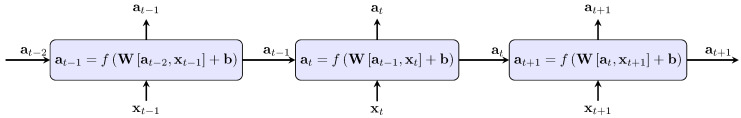
Unfolded view on a recurrent neural network. The neural network is extended by an additional loop which allows the use of previous information for the subsequent prediction.

**Figure 4 sensors-20-04581-f004:**
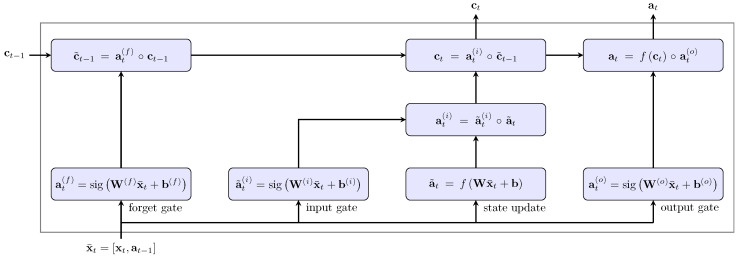
Overview of an LSTM cell. In each cell, four different networks are trained (bottom line) to be able to memorise relevant information over long time. According to the information, the internal cell state $c_{t}$ and the activation $a_{t}$ are updated and used as further inputs during the next recurrence.

**Figure 5 sensors-20-04581-f005:**
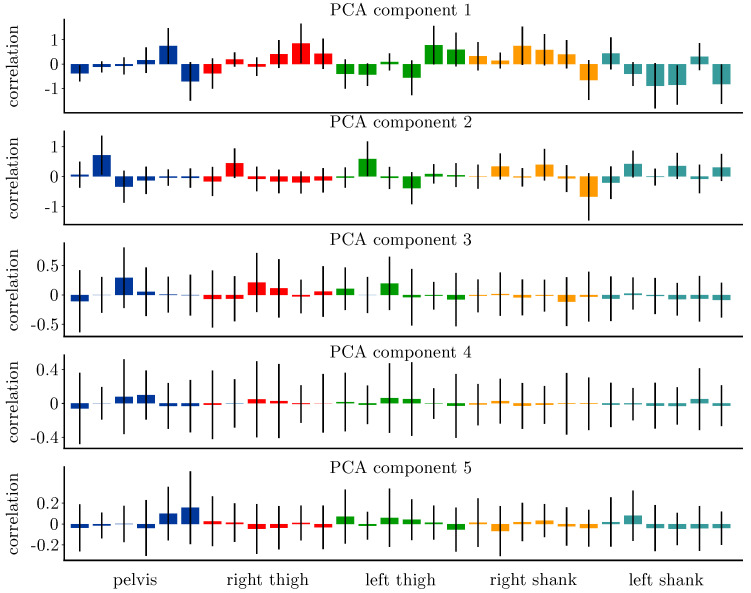
Correlation of each sensor’s reading ($acc_{x}$, $acc_{y}$, $acc_{z}$, $\omega_{x}$, $\omega_{y}$, $\omega_{z}$) with the first five PCA components. The correlation strength of the acceleration is, in general, less than that the correlation of the angular rate with the single components. The pelvis and shank sensors show stronger correlations with the PCA components than the thigh sensors.

**Figure 6 sensors-20-04581-f006:**
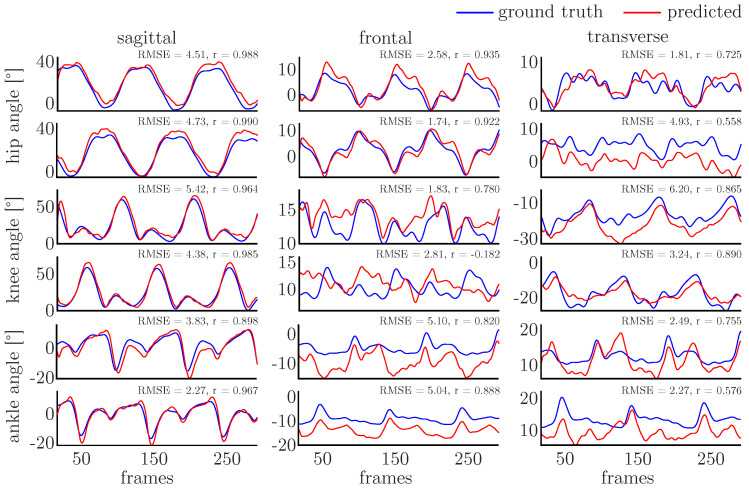
A single example of the joint angle prediction using PCA-net. The joint angles predicted for the right are displayed on top of those predicted for the left. The sample depicts the one with the median RMSE value.

**Figure 7 sensors-20-04581-f007:**
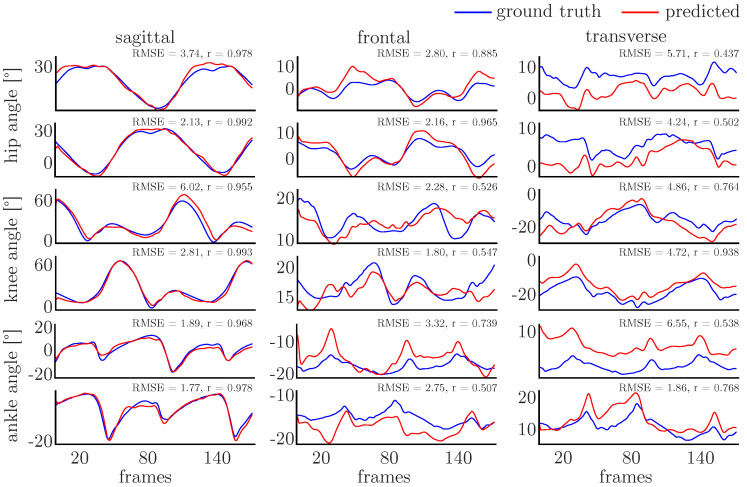
A single example of the joint angle prediction using PS-net. The joint angles predicted for the right are displayed on top of those predicted for the left. The sample depicts the one with the median RMSE value.

**Figure 8 sensors-20-04581-f008:**
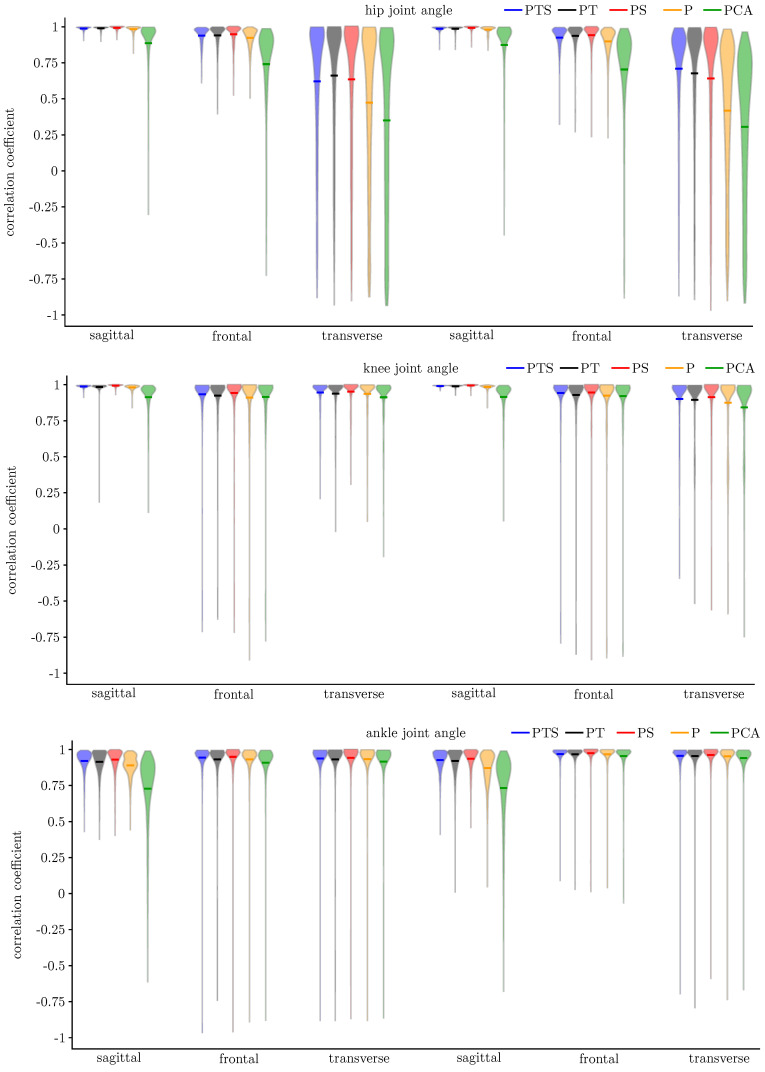
Distribution of the correlation coefficient for the (**Top**) Hip, (**Middle**) Knee and (**Bottom**) Ankle joint angle prediction.

**Table 1 sensors-20-04581-t001:** Split of the dataset.

CV	No. of Training Subjects (Samples)	No. of Validation Subjects (Samples)	No. of Test Subjects (Samples)
1	73 (56,764)	15 (14,398)	27 (16,905)
2	70 (60,145)	18 (11,017)	27 (16,905)
3	71 (59,823)	17 (11,339)	27 (16,905)
4	73 (57,270)	15 (13,892)	27 (16,905)
5	73 (58,190)	15 (12,972)	27 (16,905)

**Table 2 sensors-20-04581-t002:** Mean (standard deviation) of the correlation coefficient for the five-fold cross-validation using the different input parameters. The joint angles predicted for the left and right have been summarised.

		PCA-Net	PTS-Net	PT-Net	PS-Net	P-Net
hip	sagittal	0.891 (0.047)	0.985 (0.006)	0.988 (0.007)	0.989 (0.006)	0.979 (0.010)
	frontal	0.747 (0.104)	0.935 (0.034)	0.938 (0.031)	0.942 (0.032)	0.908 (0.037)
	transverse	0.408 (0.229)	0.677 (0.207)	0.653 (0.190)	0.646 (0.188)	0.459 (0.238)
knee	sagittal	0.925 (0.032)	0.992 (0.003)	0.990 (0.004)	0.993 (0.003)	0.978 (0.007)
	frontal	0.934 (0.043)	0.947 (0.036)	0.940 (0.039)	0.950 (0.032)	0.928 (0.052)
	transverse	0.894 (0.072)	0.929 (0.059)	0.926 (0.057)	0.936 (0.054)	0.913 (0.063)
ankle	sagittal	0.760 (0.088)	0.927 (0.035)	0.920 (0.040)	0.937 (0.033)	0.873 (0.050)
	frontal	0.941 (0.023)	0.952 (0.041)	0.956 (0.022)	0.965 (0.018)	0.938 (0.036)
	transverse	0.934 (0.036)	0.947 (0.041)	0.951 (0.029)	0.958 (0.028)	0.939 (0.033)

**Table 3 sensors-20-04581-t003:** Mean (standard deviation) of the RMSE for the five-fold cross-validation using the different input parameters. The joint angles predicted for the left and right have been summarised.

		PCA-Net	PTS-Net	PT-Net	PS-Net	P-Net
hip	sagittal	4.11 (0.96)	1.74 (0.54)	1.70 (0.58)	1.62 (0.55)	2.31 (0.62)
	frontal	1.67 (0.38)	0.95 (0.28)	0.91 (0.28)	0.87 (0.30)	1.16 (0.26)
	transverse	2.77 (0.87)	2.13 (0.86)	2.13 (0.91)	2.12 (0.92)	2.72 (0.97)
knee	sagittal	4.60 (1.06)	1.77 (0.38)	1.98 (0.51)	1.69 (0.44)	2.97 (0.55)
	frontal	2.16 (0.71)	1.58 (0.66)	1.77 (0.82)	1.54 (0.72)	2.18 (0.85)
	transverse	3.49 (1.10)	2.62 (1.06)	2.85 (1.25)	2.48 (1.09)	3.36 (1.27)
ankle	sagittal	2.49 (0.31)	1.50 (0.36)	1.58 (0.43)	1.35 (0.37)	2.01 (0.35)
	frontal	2.17 (0.56)	1.71 (0.72)	1.76 (0.72)	1.51 (0.62)	2.21 (0.72)
	transverse	1.79 (0.38)	1.39 (0.47)	1.39 (0.48)	1.21 (0.41)	1.68 (0.47)

**Table 4 sensors-20-04581-t004:** Results of the RMSE and the correlation coefficient *r* achieved in [29].

		FFNN		LSTM		PS-Net	
		RMSE	*r*	RMSE	*r*	RMSE	*r*
	sagittal	1.31	0.999	1.74	0.997	1.62	0.989
hip	frontal	1.25	0.980	1.30	0.965	0.87	0.942
	transverse	2.48	0.864	2.70	0.889	2.12	0.646
	sagittal	1.37	0.997	1.92	0.997	1.69	0.993
knee	frontal	1.55	0.793	1.92	0.681	1.54	0.950
	transverse	1.74	0.957	3.73	0.945	2.48	0.936
	sagittal	1.56	0.983	1.80	0.983	1.35	0.937
ankle	frontal	1.31	0.892	1.35	0.912	1.51	0.965
	transverse	1.76	0.891	2.14	0.920	1.21	0.958

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
