# Peer review of "Artificial Neural Networks in Motion Analysis—Applications of Unsupervised and Heuristic Feature Selection Techniques"

_sensors, 2020, doi:10.3390/s20164581_

Round 1
Reviewer 1 Report
Authors have presented a Neural networks approach in order to estimate joint
4 angles from a sparse data set, which enables the reduction of sensors necessary for the determination 5 of all three dimensional lower limb joint angles. This approach is interesting due to the possibility of optimizing the number of sensors required in real applications.
My recommendations to improve this contribution are listed below:
- In the introduction section is more appropriate to divide it into 5 subsections as follows: i) General context, ii) Motivation, iii) State-of-the-art, iv) Contribution and scope, and v) Organization of the document. With these divisions is very easy for readers to understand the main aspects of the article by simply reading the introduction section.
- In the case of mathematical nomenclature, why using superindex t previous to the variable, i.e., {t-1}^x? it is typically used as x^{t-1}.
- Why do authors decide to use the first five principal components? are the remainder of features irrelevant?
- It is possible to have a performance index that relates the number of sensors with the accuracy of the estimation of the angle. This is important since reducing the number of sensors allows reducing costs, nevertheless, it also implies errors in the final estimation.
- In conclusion, authors must be presented some future works derived from this current research.
Author Response
Please find the responses to the reviewer attached as pdf.

Reviewer 2 Report
This paper presents interesting results assessing the ability of simulated inertial sensor data to estimate joint angles. While overall I found the paper to be good and well written, there are a few key areas that need to be further discussed or strengthened.
1) The use of simulated inertial sensor data needs to be more fully explained, particularly when discussing the limitations of this approach. The authors put simulated in parenthesis when discussing this aspect of the paper. This reads as an attempt to downplay this aspect of the paper, which seems inappropriate. I would recommend removing these parenthesis. Additionally, the limitations of using this method should be more fully explained. For example, from the images in Figure 1, it appears that the simulated inertial sensors were affixed to bone. This is not realistic and eliminates noise due to muscle and skin artifacts that would occur with real, non-simulated sensors. This and other limitations should be more fully discussed in the brief paragraph focused on this in the discussion section.
2) There seems to be quite a bit of focus on accuracy levels, predominantly assessed through RMSE in this paper. I appreciate this focus on accuracy and understand its importance. However, in the Introduction and the Discussion section, the authors do not indicate the level of accuracy that would be needed for the joint angle outputs to be useful in sports medicine or clinical application. Therefore, it is unclear whether their earlier work was already 'good enough' or whether this work is 'good enough' for these application areas. Similarly, in the Introduction section, there is no indication of the RMSE or accuracy levels achieved in joint angle prediction without the use of NN or feature selection. Therefore, it is difficult to gauge the level of improvement made in performance by using NN and feature selection. Similarly, in the discussion section, the only comparative accuracy values provided are for the authors' earlier work. So, it is difficult to assess the relative accuracy of this work compared to that being done by other research groups. This made it difficult to interpret the results of this paper in terms of their potential impact and importance.
3) Additionally, in the Introduction section, the authors seem to switch between discussing papers focused on the prediction of kinetic parameters like ground reaction forces and kinematic parameters like joint angles. While I appreciate the connection between these, if you are focusing on joint angles, this should be the focus of the Introduction. If both joint angles and ground reaction forces are important in this area, why are you focusing solely on joint angles?
4) Additionally methodological information is needed concerning sample size, training vs testing data sets, and other methodological details:
-It is not clear how many participants were included in this study. What is the n? What is the mean age? What is the gender breakdown?
-Additionally, what frequency was the motion capture data collected at? You provide a range of 100 to 125 Hz, but what was actually used? Additionally, when the inertial sensor data was simulated, what frequency was included in this simulated data? Was it down-sampled? It would be rare to see a study using actual inertial sensor data with a frequency as high as 125 Hz due to the implications for data storage. Finally, in terms of the simulation, what outputs were simulated (e.g., tri-axial accelerometer, tri-axial gyroscope, etc.)? This information may be covered in reference [11], but the key information should be provided here. (I shouldn't have to read another paper to understand your paper.)
-How were the training and test data separated from each other? Did data from one participant appear in both the training and test data sets? Was separation of training and test data done for both feature selection and NN training?
5) Author contributions was not completed and is still in its generic form.
Author Response

(The authors gave the same response as above.)

Reviewer 3 Report
The subject of paper is interesting, but the current version needs significant improvement in experiments and discussion.
1) The basis of parameter setting is not clear. For example, medio-lateral (z) axis (r=-0.716) and vertical (y) axis (r=0.746), thigh sensors’ vertical (y) axes (right r = 0.847, left r=0.777), both shank sensors’ medio-lateral (z) acceleration (right r=0.749, left r=-0.896) (lines 151-153). The reason for the parameter setting should be explained.
2) In figure 6, the frontal joint angle prediction is very poor. Why the true value deviates far from the predicted value (RMSE = 2.81, r = -0.182)?
3) “Table 1. Mean (standard deviation) of the correlation coefficient for the five-fold cross-validation for all 18 joint angles estimated using the different input parameters.”
Why are there 18 joint angles in Table 1 and Table 2? I only can find 9 angles (hip-sagittal/ frontal/ transverse, knee-sagittal/ frontal/ transverse, ankle-sagittal/ frontal/ transverse).
4) The focus of this article is the application of unsupervised and infantile feature selection techniques in motion analysis. However, there have been too few experiments, analyses and discussions in this area. It is recommended that the necessary analysis and studies be added.
5) The experimental discussion on neural networks is too limited, and it is suggested to supplement the necessary data description, such as the number of test samples and training samples, the speed of prediction, etc.
Author Response

(The authors gave the same response as above.)

Round 2
Reviewer 3 Report
Accept in present form